# RNA m6A Modification Changes in Postmortem Nucleus Accumbens of Subjects with Alcohol Use Disorder: A Pilot Study

**DOI:** 10.3390/genes13060958

**Published:** 2022-05-27

**Authors:** Ying Liu, Huiping Zhang

**Affiliations:** 1Department of Psychiatry, Boston University School of Medicine, Boston, MA 02118, USA; yliu8@bu.edu; 2Department of Medicine (Biomedical Genetics), Boston University School of Medicine, Boston, MA 02118, USA

**Keywords:** alcohol use disorder, human postmortem nucleus accumbens, m6A epitranscriptome microarray, differentially methylated RNAs, functional annotation

## Abstract

Background: The nucleus accumbens (NAc) is a key brain structure mediating the rewarding effect of alcohol and drug abuse. Chronic alcohol consumption may alter RNA methylome (or epitranscriptome) in the NAc, leading to altered gene expression and thus behavioral neuroadaptation to alcohol. Methods: This pilot study profiled the epitranscriptomes of mRNAs, long noncoding RNAs (lncRNAs), and microRNAs (miRNAs) in postmortem NAc of three male Caucasian subjects with alcohol use disorder (AUD) and three matched male Caucasian control subjects using Arraystar’s m6A-mRNA&lncRNA Epitranscriptomic Microarray assay. Differentially methylated (DM) RNAs and the function of DM RNAs were analyzed by biostatistics and bioinformatics programs. Results: 26 mRNAs were hypermethylated and three mRNAs were hypomethylated in the NAc of AUD subjects (≥2-fold changes and *p* ≤ 0.05). Most of these 29 DM mRNAs are involved in immune-related pathways (e.g., IL-17 signaling). Moreover, four lncRNAs were hypermethylated and one lncRNA was hypomethylated in the NAc of AUD subjects (≥2-fold changes and *p* ≤ 0.05). Additionally, three miRNAs were hypermethylated in the NAc of AUD subjects (≥2-fold changes and *p* ≤ 0.05). Conclusions: This study revealed RNA methylomic changes in the NAc of AUD subjects, suggesting that chronic alcohol consumption may lead to AUD through epitranscriptomic RNA modifications. Our findings need to be replicated in a larger sample.

## 1. Introduction

Alcohol use disorder (AUD) affects about 4.2% (or 14.1 million) of American adults each year [1] and causes substantial morbidity and mortality. The typical symptoms of AUD include compulsive alcohol use, loss of control over drinking, and alcohol withdrawal symptoms. Twin and adoption studies have demonstrated that AUD is about 50% heritable [2]. Besides genetic variation, environmental factors, psychosocial stressors, and chronic alcohol consumption can also lead to AUD *via* epigenetic modifications [3], such as DNA methylation (as in our previous studies [4,5,6,7,8,9,10,11,12]) and histone modifications [13] at the transcriptional level.

Epigenetic modifications can also occur at the posttranscriptional level. Among over 170 different types of RNA modifications identified so far [14], the methylation at N-6 adenosine (m6A) in RNAs has attracted much attention because m6A is the most common internal modification in eukaryotic RNAs [15]. m6A sites are enriched near stop codons, in 3′ UTRs, in long conserved internal exons of mRNAs [16,17], and in the last exons of lncRNAs [16], thus regulating transcript splicing, stability, and translation as well as microRNA binding. m6A is dynamic and responds to a variety of stimuli, thus translating stimulatory signals into cellular activity by influencing gene expression [16]. RNA methylomic changes can cause a number of brain disorders, such as microcephaly [18], epilepsy [19], intellectual disability [20], and depression [21]. RNA epigenetic modifications in specific brain regions may also increase the risk of developing neuropsychiatric disorders, including AUD.

The nucleus accumbens (NAc) plays a central role in the mesolimbic reward pathway. It mediates the process of motivation and emotion as well as the rewarding effect of alcohol and drugs of abuse [22]. There is a large body of evidence supporting the effect of alcohol on the NAc as well as the role of the NAc in developing AUD. Alcohol consumption enhances extracellular dopamine levels in the NAc, leading to subjective feelings of euphoria and stimulation and thus creating a craving for drinking more alcohol [23,24].

To identify alcohol-responsive genes expressed in the NAc, Flatscher-Bader et al. performed cDNA microarray analysis of the transcriptome in postmortem NAc and found that alcohol-responsive genes were associated with vesicle formation and regulation of cell architecture, suggesting a neuroadaptation to chronic alcohol exposure at the level of synaptic structure and function [25]. In our recent study with the use of RNA sequencing (RNA-seq) to profile mRNA and microRNA (miRNA) transcriptomic changes in postmortem NAc of AUD subjects, we unraveled AUD-associated mRNA-miRNA pairs and their potentially influenced pathways (such as the CREB signaling in neurons) [26]. Recently, Drake et al. assessed the role of long-noncoding RNA (lncRNAs) in postmortem NAc of AUD subjects [27]. To understand the epigenetic mechanisms behind alcohol-induced neuroadaptative changes in the NAc, Cervera-Juanes et al. used rhesus macaques as models to identify NAc DNA methylation signals that distinguished alcohol-naive (AN), low/binge (L/BD), and heavy/very heavy (H/VHD) drinking primates [28].

To date, no study is known to have examined RNA methylomic (or epitranscriptomic) changes in AUD subjects. We hypothesize that chronic alcohol use could alter methylation levels of both coding and noncoding RNAs that are responsive to alcohol and expressed in reward-related brain regions (particularly the NAc). In this pilot study, we used the RNA methylation microarray assay to profile mRNA/lncRNA/miRNA methylation changes in postmortem NAc of AUD subjects.

## 2. Materials and Methods

### 2.1. Human Postmortem NAc Tissue Samples

Freshly frozen autopsy brain tissue samples were obtained from the New South Wales Brain Tissue Resource Centre (NSWBTRC) in Australia. They were dissected from postmortem NAc of three male AUD and three matched male control subjects. All subjects were Caucasian Australians with no history of illicit drug abuse or major psychotic disorders (such as schizophrenia and bipolar disorder) according to the criteria in the Diagnostic and Statistical Manual of Mental Disorder 4th Edition (DSM-IV) [29]. Control subjects had no history of AUD. The three AUD subjects died of heart disease, liver disease, and toxins, respectively. The death of the three control subjects was due to cardiovascular diseases. The demographic information of brain tissue samples is presented in Table 1. Except for the amount of daily alcohol consumption, other demographic variables (including age, postmortem intervals, brain weight, and brain pH) were not significantly different in their measurements between cases and controls. The study was approved by the Boston University Medical Campus Institutional Review Board (IRB approval numbers: H-41895).

### 2.2. Total RNA Isolation from Postmortem NAc Tissue Samples

Total RNAs were isolated from 10–50 mg of postmortem NAc tissue samples using the miRNeasy Mini Kit (QIAGEN, Valencia, CA, USA). RNA integrity number (RIN) and concentration were measured using the Agilent 2100 Bioanalyser with the Agilent RNA 6000 Nano Kit (Agilent Technologies, Santa Clara, CA, USA). The mean RIN of RNAs isolated from AUD NAc tissue samples was 6.1 (±0.5), and the mean RIN of RNAs isolated from control NAc tissue samples was 7.4 (±2.4). Denaturing agarose gel electrophoresis showed good quality of RNAs extracted from both the case and control NAC tissue samples (Figure 1). The 28S and 18S ribosomal RNA (rRNA) bands are sharp and intense. Genomic DNA contamination of the RNA preparation was not observed (i.e., no high molecular weight smear or bands migrating above the 28S rRNA band).

### 2.3. RNA Methylome Profiled by Arraystar m6A-mRNA&lncRNA Epitranscriptomic Microarray Assay

#### 2.3.1. m6A Immunoprecipitation (IP)

1–3 μg total RNA and m6A spike-in control mixture were added to 300 μL of IP buffer (50 mM Tris-HCl, pH7.4, 150 mM NaCl, 0.1% NP40, 40 U/μL RNase Inhibitor) containing 2 μg of anti-m6A rabbit polyclonal antibody (Synaptic Systems, Goettingen, Germany). The reaction was incubated with head-over-tail rotation at 4 °C for 2 h. 20 μL of Dynabeads™ M-280 Sheep Anti-Rabbit IgG suspension (Invitrogen, Waltham, MA, USA) per sample was blocked with freshly prepared 0.5% bovine serum albumin (BSA) at 4 °C for 2 h, washed three times with 300 μL of IP buffer, and resuspended in the total RNA-antibody mixture prepared above. The RNA binding to the m6A-antibody beads was carried out with head-over-tail rotation at 4 °C for 2 h. The beads were then washed three times with 500 μL of IP buffer and twice with 500 μL Wash buffer (50 mM Tris-HCl, pH7.4, 50 mM NaCl, 0.1% NP40, 40 U/μL RNase Inhibitor). The enriched RNA was eluted with 200 μL Elution buffer (10 mM Tris-HCl, pH7.4, 1 mM EDTA, 0.05% SDS, 40 U Proteinase K) at 50 °C for 1 h. The RNA was extracted by acid phenol-chloroform and ethanol precipitated.

#### 2.3.2. Labeling and Hybridization

“IP” RNAs and “Sup” RNAs were added with an equal amount of calibration spike-in control RNA, separately amplified, and labeled with Cy3 (for “Sup”) and Cy5 (for “IP”) using the Arraystar Super RNA Labeling Kit (Rockville, MD, USA). The synthesized cRNAs were purified by the RNeasy Mini Kit (Hilden, German). The concentration and specific activity (pmol dye/μg cRNA) were measured with NanoDrop ND-1000 (Thermo Fisher Scientific, Waltham, MA, USA). 2.5 μg of Cy3 and Cy5 labeled cRNAs were mixed. The cRNA mixture was fragmented by adding 5 μL of Blocking Agent and 1 μL of 25× Fragmentation Buffer, heated at 60 °C for 30 min, and combined with 25 μL of 2× Hybridization buffer. 50 μL of hybridization solution was dispensed into the gasket slide and assembled to the m6A-mRNA&lncRNA Epitranscriptomic Microarray slide. The slides were incubated at 65 °C for 17 h in an Agilent Hybridization Oven (Agilent Technologies, Santa Clara, CA, USA). The hybridized arrays were washed, fixed, and scanned using an Agilent Scanner G2505C (Agilent Technologies, Santa Clara, CA, USA).

#### 2.3.3. RNA Methylation Array Data Processing

The Agilent Feature Extraction software (version 11.0.1.1) was used to analyze acquired array images. Raw intensities of IP (immunoprecipitated, Cy5-labelled) and Sup (supernatant, Cy3-labelled) were normalized with an average of log_2_-scaled Spike-in RNA intensities. After Spike-in normalization, the probe signals having Present (P) or Marginal (M) QC flags in at least 1 out of 6 samples was retained as “All Targets Value” in the Excel sheet for further “m6A methylation level” analyses. The “m6A methylation level” for a transcript was calculated as the percentage of modified RNA (%Modified) in all RNAs based on the IP (Cy5-labelled) and Sup (Cy3-labelled) normalized intensities. Raw intensities of IP (immunoprecipitated, Cy5-labelled) and Sup (supernatant, Cy3-labelled) were normalized with the average of log_2_-scaled Spike-in RNA intensities.
%Modified = Modified RNA/Total RNA = *IP*/(*IP* + *Sup*) = *IP*_cy5 normalized intensity_/(*IP*_cy5 normalized intensity_ + *IP*_cy3 normalized intensity_)log_2_(*IP*_cy5 normalized intensity_) = log_2_(*IP*_Cy5 raw_) − Average[log_2_(*IP*_spike-in_Cy5 raw_)]log_2_(*Sup*_cy3 normalized intensity_) = log_2_(*Sup*_Cy3 raw_) − Average[log_2_(*Sup*_spike-in_Cy3 raw_)]

### 2.4. Analysis of Differentially Methylated m6A Sites

A t-test was used to compare two groups (i.e., disease vs. control) for differential m6A modification and calculate the fold change (FC) and statistical significance of the difference (*p*-value) for each transcript. The threshold for statistical significance was set at |FC| ≥ 2 and *p*-value < 0.05.

### 2.5. Hierarchical Clustering Using Heatmaps

Hierarchical clustering was performed using the R software. It arranged samples together based on the similarities of their m6A methylation level (or quantity) and the closeness of their relationships as displayed in the dendrogram on top of the heatmaps.

### 2.6. Gene Ontology (GO) and Pathway Analysis of Differentially m6A-Methylated mRNAs

The web-based gene set enrichment analysis tool Enrichr [30] was used to perform the Gene Ontology (GO) and pathway analysis. The GO and pathway analysis associated the differentially m6A-methylated mRNAs to certain gene ontological functions (or GO terms) [Biological Process (BP), Cellular Component (CC) and Molecular Function (MF)] and Kyoto Encyclopedia of Genes and Genomes (KEGG) pathways. The statistical significance of the enrichment was calculated by Fisher Exact tests. The *p*-value denotes the significance of GO terms or pathways enriched in differentially methylated mRNAs. The lower the *p*-value, the more significant the GO term or pathway.

## 3. Results

### 3.1. Differentially m6A-Methylated mRNAs, lncRNAs, and microRNAs

The Arraystar m6A-mRNA&lncRNA Epitranscriptomic Microarray assay determined methylation levels of 36,790 mRNAs, 8540 lncRNAs, and 2377 other small noncoding RNAs. By the default differential methylation threshold (|FC| ≥ 2 and *p*-value < 0.05), 26 mRNAs were hypermethylated, while three mRNAs were hypomethylated in the NAc of AUD subjects (Figure 2a). Moreover, four lncRNAs were hypermethylated, while one lncRNA was hypomethylated in the NAc of AUD subjects (Figure 2b). Additionally, three miRNAs were hypermethylated in the NAc of AUD subjects (Figure 2c). Differentially methylated mRNAs, lncRNAs, and miRNAs are listed in Table 2. Heatmaps were used to visualize differentially methylated mRNAs, lncRNAs, and miRNAs and hierarchically cluster RNAs (rows) and tissues (columns) (Figure 3).

### 3.2. Gene Ontology (GO) Enrichment Analysis Results

The top 18 Gene Ontology (GO) enrichment terms (adjusted *p* < 0.05) are shown in Figure 4. Among them, three were significant biological process (BP) terms, including *Positive Regulation of Acute Inflammatory Response* (GO: 0002675; adjusted *p* = 4.5 × 10^−4^), *Regulation of Neuroinflammatory Response* (GO: 0150077; adjusted *p* = 0.046), and *Regulation of Acute Inflammatory Response* (GO: 0002673; adjusted *p* = 0.046). Other 15 were significant molecular function (MF) terms, including *N-acetylgalactosamine 4-O-sulfotransferase activity* (GO: 0001537; adjusted *p* = 0.048), *Anion:Sodium Symporter Activity* (GO: 0015373; adjusted *p* = 0.048), *Oncostatin-M Receptor Activity* (GO: 0004924; adjusted *p* = 0.048), *Ubiquinol-cytochrome-c Reductase Activity* (GO: 0008121; adjusted *p* = 0.048), *Amidine-lyase Activity* (GO: 0016842; adjusted *p* = 0.048), *Leukemia Inhibitory Factor Receptor Activity* (GO: 0004923; adjusted *p* = 0.048), *Cytokine Receptor Binding* (GO: 0005126; adjusted *p* = 0.048), *Interleukin-6 Receptor Binding* (GO: 0005138; adjusted *p* = 0.048), *Phosphatidylinositol-4,5-bisphosphate 5-phosphatase Activity* (GO: 0004439; adjusted *p* = 0.048), *Ciliary Neurotrophic Factor Receptor Activity* (GO: 0004897; adjusted *p* = 0.048), *Phosphatidylinositol Phosphate 5-phosphatase Activity* (GO: 0034595; adjusted *p* = 0.048), *Ciliary Neurotrophic Factor Receptor Binding* (GO: 0005127; adjusted *p* = 0.048), *Potassium:Chloride Symporter Activity* (GO: 0015379; adjusted *p* = 0.048), *Sodium:Chloride Symporter Activity* (GO: 0015378; adjusted *p* = 0.048), and *Phosphatidylinositol-4,5-bisphosphate Phosphatase Activity* (GO: 0106019; adjusted *p* = 0.048). No significant cellular component (CC) terms with adjusted *p* < 0.05 were identified.

### 3.3. KEGG Pathway Enrichment Analysis Results

None of the KEGG pathways reached the adjusted significance threshold (adjusted *p* < 0.05). The top 10 KEGG enrichment terms are shown in Figure 5. The top 10 KEGG pathways (with unadjusted *p* ≤ 0.030 and adjusted *p* ≤ 0.200) included: *IL-17 Signaling Pathway* (unadjusted *p* = 3.4 × 10^−4^), *Inflammatory Bowel Disease* (unadjusted *p* = 0.004), *Cytokine-cytokine Receptor Interaction* (unadjusted *p* = 0.009), *C-type Lectin Receptor Signaling Pathway* (unadjusted *p* = 0.010), *Th17 Cell Differentiation* (unadjusted *p* = 0.010), *TNF Signaling Pathway* (unadjusted *p* = 0.011), *Alzheimer’s Disease* (unadjusted *p* = 0.016), *Non-alcoholic Fatty Liver Disease* (unadjusted *p* = 0.021), *JAK-STAT Signaling Pathway* (unadjusted *p* = 0.023), and *Nitrogen Metabolism* (unadjusted *p* = 0.024).

## 4. Discussion

This pilot study explored the epitranscriptomic (or RNA methylomic) changes in postmortem NAc of male AUD subjects. We identified 29 mRNAs, 5 lncRNAs, and 3 miRNAs that were differentially methylated (|FC| ≥ 2 and *p* < 0.05) in AUD subjects. The top GO terms and KEGG pathways in which the above 29 differentially methylated mRNAs were highly enriched included inflammatory or immune response. In other words, chronic alcohol consumption may alter the mRNA methylation status of inflammation or immune response-related genes.

Accumulating evidence suggests that acute alcohol exposure may lead to anti-inflammatory responses of the immune system [31], while chronic exposure may result in pro-inflammatory reactions that remain present during abstinence [32]. It is also known that circulating cytokines can mediate the gut–brain communication. Cytokines can cross the blood–brain barrier to enter the cerebrospinal fluid and the brain and thus induce neuroinflammation and result in altered mood, cognition, and drinking behavior. In this pilot study, we demonstrated that AUD-associated and differentially methylated mRNAs were likely involved in inflammation or immune response (such as IL-17 signaling, inflammatory bowel disease, and cytokine–cytokine receptor interaction) (Figure 5).

Alcohol use or exposure can alter genome-wide expression levels of genes including those genes involved in inflammation or immune response. Using rats as models, Sanchez-Marin et al. demonstrated that ethanol exposure caused a higher expression level of neuroinflammatory-associated genes in the rat brain [33]. Alcohol may enhance the expression of neuroinflammatory-related genes through epigenetic mechanisms such as RNA methylation. To understand whether the expression of the above 29 differentially methylated mRNAs was also significantly altered in postmortem NAc of AUD subjects, we re-analyzed our RNA-seq data [26] by comparing mRNA transcriptome profiles of six male Caucasian AUD subjects and six male Caucasian male control subjects (including the 3 male cases and the 3 male controls for the present study). However, we did not observe significant expression changes of the above 29 differentially methylated mRNAs at the level of expression fold changes over two and *p* values less than 0.05. Nevertheless, we cannot exclude the possibility that the methylation alterations in these 29 mRNAs could influence their expression on a smaller scale.

This pilot study is limited in several ways. First, it was mainly limited by its small sample size. We expect that more differentially methylated mRNAs could be identified when the sample size is larger and a logistic regression model (with confounding factors being considered) is used. Second, this pilot study only analyzed the epitranscriptomic changes in the NAc of AUD subjects. Since several other critical brain regions (such as the prefrontal cortex and the ventral tegmental area) are also components of the brain reward center, AUD-associated epitranscriptomic changes in other brain regions should also be examined. Third, since both sex-specific gene expression [34] and sex-specific DNA methylation [8] changes were observed in postmortem brains of AUD subjects in our previous studies, we also expect that sex-specific RNA methylation changes should be revealed in the brain of AUD subjects when a larger brain tissue sample is available for the epitranscriptome study. Fourth, this pilot study is also limited by analyzing AUD-associated brain RNA methylation changes in only the Caucasian population. Our future studies should also investigate AUD-associated brain RNA methylation changes in our populations since RNA methylation patterns can be population-specific. Finally, the functional role of RNA methylation in regulating RNA expression and protein translation should be investigated by innovative approaches such as the CRISPR-based m6A editing technology [35].

## 5. Conclusions

This pilot work is the first step to the further examination of RNA methylomic variation in the brain of subjects affected with AUD in order to understand the epitranscriptomic mechanisms of AUD. Given that the methylation status of mRNAs involved in inflammation or immune response was altered in the brain of AUD subjects, future studies may adopt novel approaches to modify the methylation of these mRNAs to reduce systemic inflammation and thus benefit AUD treatment outcomes and reduce alcohol relapse.

## Figures and Tables

**Figure 1 genes-13-00958-f001:**
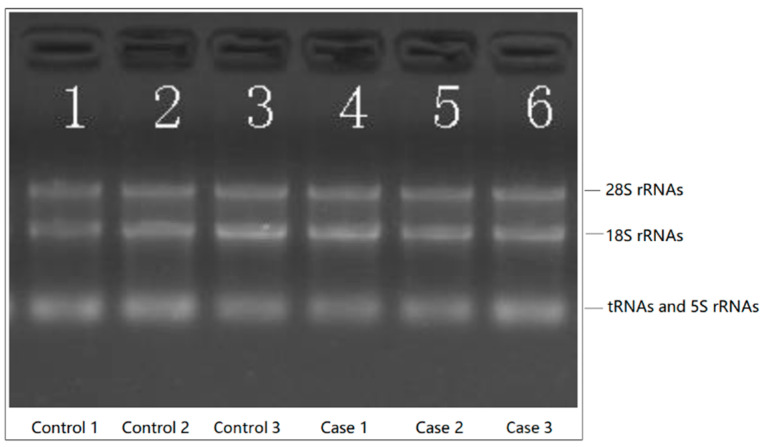
RNA integrity and genomic DNA contamination examined by denaturing agarose gel electrophoresis.

**Figure 2 genes-13-00958-f002:**
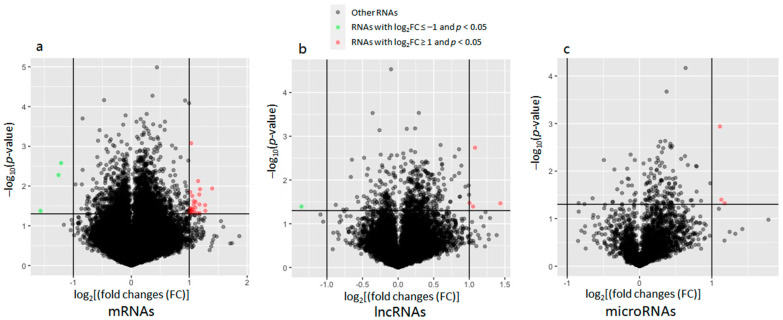
Volcano plots displaying differentially methylated RNAs (mRNAs, lncRNAs, and miRNAs) in postmortem nucleus accumbens (NAc) of subjects with alcohol use disorder (AUD). The vertical axis (y-axis) corresponds to the negative log_10_ of the *p*-value, and the horizontal axis (x-axis) displays the log_2_ of fold changes (FC). The red dots represent up-regulated RNAs (log_2_FC > 1.0 and *p* < 0.05) and the green dots represent downregulated RNAs (log_2_FC < −1.0 and *p* < 0.05). The horizontal line shows the *p*-value cutoff (*p* = 0.05) with points above the line having the *p*-value < 0.05 and points below the line having the *p*-value > 0.05. The two vertical lines indicate 2-fold changes.

**Figure 3 genes-13-00958-f003:**
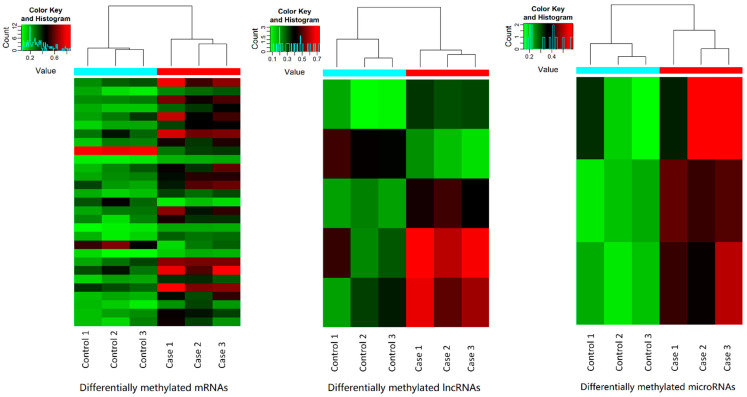
Heatmaps visualing differentially methylated mRNAs (*n* = 29), lncRNAs (*n* = 5), and miRNAs (*n* = 3). The red–green gradient color scheme represents the high and low m6A-methylation levels as referenced in the Color Key. The top dendrogram shows the closeness among the samples. The sample group membership is indicated by the color bars above the heat map.

**Figure 4 genes-13-00958-f004:**
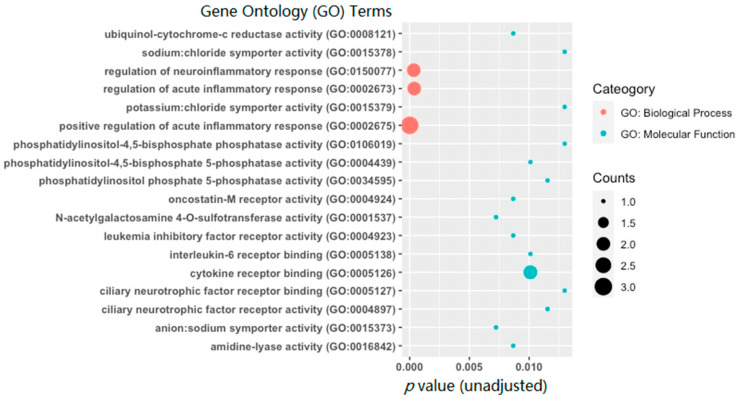
A bubble plot of the top 18 Gene Ontology (GO) terms (adjusted *p* < 0.05) obtained by enrichment analysis of differentially methylated mRNAs (|FC| ≥ 2 and *p*-value < 0.05). GO terms are color-coded by subcategory (biological process vs. molecular function). The size of the bubble for each enrichment term corresponds to the number of enriched genes within that GO term.

**Figure 5 genes-13-00958-f005:**
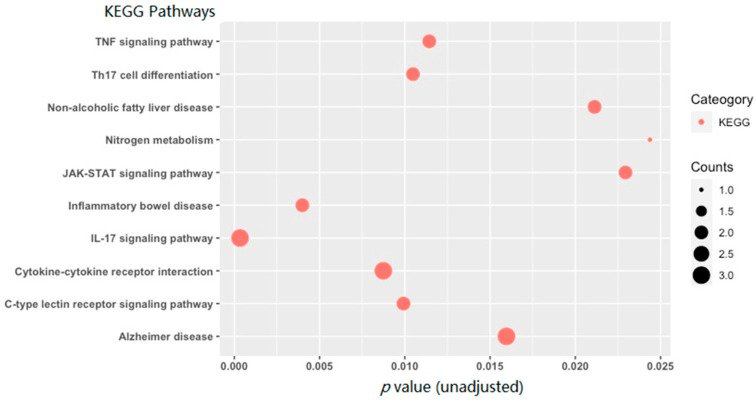
A bubble plot of the top 10 KEGG pathways (unadjusted *p* ≤ 0.03 and adjusted *p* ≤ 0.200) obtained by enrichment analysis of differentially methylated mRNAs (|FC| ≥ 2 and *p*-value < 0.05). The size of the bubble for each enrichment term corresponds to the number of enriched genes within that KEGG term.

**Table 1 genes-13-00958-t001:** Demographic information of postmortem nucleus accumbens (NAc) tissue samples.

	AUD Cases (*n* = 3)	Controls (*n* = 3)	*t*-Tests
Daily Alcohol Use in grams (Mean ± SD)	124.7 (±45.8)	11.3 (±4.9)	t = 3.24, *p* = 0.035
Age, Year (Mean ± SD)	49.3 (±5.6)	48.3 (±5.7)	t = 0.16, *p* = 0.880
Postmortem interval in hours (Mean ± SD)	43.2 (±5.6)	35.0 (±16.0)	t = 0.61, *p* = 0.575
Brain weight in grams (mean ± SD)	1362.7 (±127.1)	1611.7 (±87.8)	t= −2.04, *p* = 0.110
Brain pH (mean ± SD)	6.6 (±0.2)	6.7 (±0.2)	t = −0.64, *p* = 0.557
RNA integrity number (RIN) (Mean ± SD)	6.1 (±0.5)	7.4 (±2.4)	t = −2.95, *p* = 0.042

**Table 2 genes-13-00958-t002:** Differentially methylated mRNAs, lncRNAs, and miRNAs in postmortem NAc of AUD subject.

RNAs	Cases (Mean)	Controls (Mean)	Regulation	Log_2_FC	*p*-Value	FDR
mRNAs:						
*IL6*	0.660	0.293	up	1.17	0.016	0.726
*RNASEH2C*	0.287	0.130	up	1.14	0.049	0.726
*IL17F*	0.556	0.250	up	1.15	0.007	0.726
*ZFP1*	0.371	0.163	up	1.18	0.048	0.726
*TRIM60*	0.572	0.283	up	1.02	0.014	0.726
*KIAA0195*	0.426	0.175	up	1.28	0.042	0.726
*CA6*	0.662	0.329	up	1.01	0.038	0.726
*CATSPER4*	0.458	0.224	up	1.03	0.001	0.726
*GRAPL*	0.187	0.090	up	1.05	0.018	0.726
*ARMC4*	0.482	0.241	up	1.00	0.045	0.726
*CATG00000112871.1*	0.485	0.230	up	1.08	0.029	0.726
*P2RY2*	0.524	0.254	up	1.05	0.038	0.726
*SP8*	0.320	0.157	up	1.03	0.043	0.726
*CHST8*	0.532	0.246	up	1.11	0.025	0.726
*TMPRSS6*	0.419	0.202	up	1.05	0.037	0.726
*ARSA*	0.194	0.074	up	1.40	0.011	0.726
*CHAC1*	0.295	0.130	up	1.18	0.029	0.726
*CDK20*	0.168	0.078	up	1.11	0.035	0.726
*OR2V1*	0.623	0.274	up	1.19	0.012	0.726
*OSMR*	0.737	0.345	up	1.09	0.045	0.726
*H1FOO*	0.359	0.178	up	1.01	0.048	0.726
*UQCRC1*	0.698	0.334	up	1.06	0.047	0.726
*CATG00000071754.1*	0.429	0.177	up	1.27	0.030	0.726
*INPP5B*	0.266	0.131	up	1.02	0.043	0.726
*SLC12A3*	0.417	0.198	up	1.08	0.024	0.726
*PSMG2*	0.358	0.177	up	1.02	0.036	0.726
*PTGS2*	0.336	0.804	down	−1.26	0.005	0.726
*TBCA*	0.121	0.359	down	−1.57	0.042	0.726
*RPL35A*	0.232	0.538	down	−1.21	0.003	0.726
lncRNAs:						
*ABCG1*	0.310	0.115	up	1.44	0.034	0.774
*AC105253.1*	0.435	0.206	up	1.08	0.002	0.774
*SIX4*	0.674	0.325	up	1.05	0.040	0.774
*RP11-121J20.1*	0.575	0.287	up	1.00	0.033	0.774
*CTD-2308N23.4*	0.165	0.424	down	−1.36	0.040	0.774
miRNAs:						
*hsa-mir-4524b*	0.498	0.220	up	1.18	0.048	0.779
*pri-3-hsa-mir-7157*	0.430	0.199	up	1.11	0.001	0.686
*hsa-mir-1273h*	0.434	0.198	up	1.13	0.040	0.779

## Data Availability

The data that support the findings of this study are available from the corresponding author upon reasonable request.

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
