# Peer review of "RNA m6A Modification Changes in Postmortem Nucleus Accumbens of Subjects with Alcohol Use Disorder: A Pilot Study"

_genes, 2022, doi:10.3390/genes13060958_

Round 1
Reviewer 1 Report
The presented manuscript "Epitranscriptomic Changes in Postmortem Nucleus Accumbens of Subjects with Alcohol Use Disorder: A Pilot Study" is largely well written and generally informative. There seems, however, to be a few minor concerns in this manuscript. No major comments regarding the method of genetic analysis. The introduction and discussion are clearly written. The authors also explained why they conducted the study on a very small group of test subjects and a control group. I get a few minor bugs.
- However, the authors did not explain why they used the t (paramedical) test for the study.
In such a small group, not all conditions could be met, e.g. normal distribution (Table 1.).
- Did not report the underlying cause of death in the Subjects with Alcohol Use Disorder group and the control group.
- In the Subjects with Alcohol Use Disorder group, they did not report how many years the addiction lasted.
Author Response
Response to Reviewer 1 Comments
Point 1: The presented manuscript "Epitranscriptomic Changes inPostmortem Nucleus Accumbens of Subjects with Alcohol UseDisorder:APilotStudy"islargelywellwrittenandgenerally Disorder: A Pilot Study” is largely well written and generally informative. There seems, however, to be a few minor concerns in this manuscript. No major comments regarding the method ofgenetic analysis. The introduction and discussion are clearly written. The authors also explained why they conducted the study on a very small group of test subjects and a control group. I get a few minor bugs.
Response 1: We thank the reviewer for the positive review of our manuscript.
Point 2: However, the authors did not explain why they used the t (paramedical) test for the study.
Response 2: Since the sample size was small (3 cases and 3 matched controls), we only performed t-tests since confounding factors (often included in regression analysis) may mask an actual association between alcohol use disorder and altered RNA methylation. In the Discussion section, we addressed this issue with the following sentence”
“We expect that more differentially methylated mRNAs could be identified when the sample size is larger and a logistic regression model (with confounding factors being considered) is used” (Page 9, Lines 300-302).
Point 3: In such a small group, not all conditions could be met, e.g. normal distribution (Table 1).
Response 3: We agree with the reviwer that a normal distribution analysis cannot be performed given the small size of the sample.
Point 4: Did not report the underlying cause of death in the Subjects with Alcohol Use Disorder group and the control group.
Response 4: The three AUD subjects died of heart disease, liver disease, and toxins, respectively. The death of the three control subjects were due to cardiovascular diseases (Page 2, Lines 86-88).
Point 5: In the Subjects with Alcohol Use Disorder group, they did not report how many years the addiction lasted.
Response 5: This information is not available from deceased patients.

Reviewer 2 Report
This is an interesting study focused on the AUD and RNA methylome in the postmortem nucleus accumbens of subjects. Following comments can be taken into consideration.
- Description the ethical approval for the barin samples in the methods section.
- Add statistic method for Table 1, maybe T test.
- Genes should be italic in Table 2 and the main text.
- What is the relationship between differentially methylated mRNA, lncRNA, and miNRA. Try to draw a figure to show the interaction.
- lncRNA belongs to the small noncoding RNA. So the title of hsa-mir-4524b, pri-3-has-mir-7157 and has-mir-1273h should be miRNA in Figure 2 and Table 2.
- Add CC of BP in Figure 4.
- Two Figure 4 in the manuscript, the GO enrichment should be Figure 5. And in the caption of this figure, p or p-value should be revised.
- Many inflammatory indicators are enriched, so what is the relationship between AUD and inflammation, this should be discussed more fully.
- In the title, the epitranscriptomic change can be revised as RNA m6A modification changes. Epitranscriptome includes several types, and this study means m6A modificaiton actually.
Author Response
Response to Reviewer 2 Comments
Point 1: This is an interesting study focused on the AUD and RNA methylome in the postmortem nucleus accumbens of subjects. Following comments can be taken into consideration.
Response 1: We thank the reviewer for the positive feedback.
Point 2: Description the ethical approval for the barin samples in the methods section.
Response 2: The study was approved by the Boston University Medical Campus Institutional Re-view Board (IRB) (Page 2, Lines 91-93).
Point 3: Add statistic method for Table 1, maybe T test.
Response 3: As suggested, we added t-test statistics in Table 1.
Point 4: Genes should be italic in Table 2 and the main text.
Response 4: Now all gene names are in italic in Table 2 and the main text.
Point 5:. What is the relationship between differentially methylated mRNAs, lncRNAs, and miRNAs. Try to draw a figure to show the interaction.
Response 5: We thank the reviewer for this thoughtful comment. We did use the Ingenuity Pathway Analysis (IPA) software to analyze the interactions of differentially methylated mRNAs, lncRNAs, and miRNAs (with p < 0.05 and fold changes >=2.0). Since the function of the lncRNAs and miRNAs are unknown, after running the analysis by IPA, we got the message “Unable to create Graphical Summary for this analysis. This occurs when the analysis does not contain enough connectable entities with sufficiently high z-score”.
Point 6:. lncRNA belongs to the small noncoding RNA. So the title of hsa-mir-4524b, pri-3-has-mir-7157 and has-mir-1273h should be miRNA in Figure 2 and Table 2.
Response 6: We thank the reviewer for pointing this out. We have changed “small noncoding RNAs” to “microRNAs” in Figure 2 and Table 2 and also in the main text.
Point 7: Add CC of BP in Figure 4.
Response 7: Since no signficantly enriched Cellular Component (CC) terms were identified, CC terms were not shown in Figure 4.
Point 8: Two Figure 4 in the manuscript, the GO enrichment should be Figure 5. And in the caption of this figure, p or p-value should be revised.
Response 8: We apologize for the typo. We have corrected it.
Point 9: Many inflammatory indicators are enriched, so what is the relationship between AUD and inflammation, this should be discussed more fully.
Response 9: We are grateful to the reviewer for this constructive comment. In fact, we have already discussed this issue in the 2nd paragraph of the Discussion section (Page 9, Lines 276-284).
Point 10: In the title, the epitranscriptomic change can be revised as RNA m6A modification changes. Epitranscriptome includes severaltypes, and this study means m6A modificaiton actually.
Response 10: We appreciate the comment, and we have changed the title of manuscript to “RNA m6A Modification Changes in Postmortem Nucleus Ac-cumbens of Subjects with Alcohol Use Disorder: A Pilot Study”.

Reviewer 3 Report
Dear authors,
First of all, I’d like to give a great congratulation to them for nice and successful study. I think that the topic and idea is novel enough to attract much interest to the readers. Also, their study was well designed, and methods were also reasonable and scientific. Their pilot study revealed RNA methylomic changes in the nucleus accumbens of alcohol use disorder subjects, suggesting that chronic alcohol consumption may lead to alcohol use disorder through epitranscriptomic RNA modifications. This manuscript can be much helpful for the readers who are doing research on epigenetics. However, it can be concerns how authors should exclude the possibility of influences by other systemic conditions in the individual subjects. Readers can be curious about the individual cause of death, because other systemic condition, such as systemic cancer or vascular disease can fully influence on the expression of epitranscriptoms in brain including nucleus accumbens. I hope that authors can explain the principles of excluding this kind of possibility.
Good luck.
Author Response
Response to Reviewer 3 Comments
Point 1: First of all, I’d like to give a great congratulation to them for niceand successful study. I think that the topic and idea is novel enough to attract much interest to the readers. Also, their study was well designed, and methods were also reasonable and scientific. Their pilot study revealed RNA methylomic changes in the nucleus accumbens of alcohol use disorder subjects, suggesting that chronic alcohol consumption may lead to alcohol use disorder through epitranscriptomic RNA modifications. This manuscript can be much helpful for the readers who are doing research on epigenetics.
Response 1: We really appreciate the postive feedback from the reviewer.
Point 2: However, it can be concerns how authors should exclude the possibility of influences by other systemic conditions in the individual subjects. Readers can be curious about the individual cause of death, because other systemic condition, such as systemic cancer or vascular disease can fully influence on the expression of epitranscriptoms in brain including nucleus accumbens. I hope that authors can explain the principles of excluding this kind of possibility.
Response 2: We thank the reviewer for this thoughtful comment. In the Materials and Methods section, we have provided the cause of death of AUD and control subjects (Page 2, Lines 86-88). We agree with the reviewer that other systemic or disease conditions (such as cancer or vascular diseases) can influence the transcriptome and epitranscriptome profiles of genes expressed in brain nucleus accumbens. In future studies, when more brain tissue samples are available, we will certainly explore this issue.
